# PARP1 Activation Induces HMGB1 Secretion Promoting Intestinal Inflammation in Mice and Human Intestinal Organoids

**DOI:** 10.3390/ijms24087096

**Published:** 2023-04-12

**Authors:** Roberta Vitali, Anna Barbara Mancuso, Francesca Palone, Claudio Pioli, Vincenzo Cesi, Anna Negroni, Salvatore Cucchiara, Salvatore Oliva, Claudia Carissimi, Ilaria Laudadio, Laura Stronati

**Affiliations:** 1Laboratory of Biomedical Technologies, Agenzia Nazionale per le Nuove Tecnologie, l’Energia e lo Sviluppo Economico Sostenibile (ENEA), 00123 Rome, Italy; 2Department of Maternal Infantile and Urological Sciences, Sapienza University, 00161 Rome, Italy; 3Department of Molecular Medicine, Sapienza University, 00161 Rome, Italy

**Keywords:** inflammation, gut, HMGB1, PARP1

## Abstract

Extracellular High-mobility group box 1 (HMGB1) contributes to the pathogenesis of inflammatory disorders, including inflammatory bowel diseases (IBD). Poly (ADP-ribose) polymerase 1 (PARP1) has been recently reported to promote HMGB1 acetylation and its secretion outside cells. In this study, the relationship between HMGB1 and PARP1 in controlling intestinal inflammation was explored. C57BL6/J wild type (WT) and PARP1^−/−^ mice were treated with DSS to induce acute colitis, or with the DSS and PARP1 inhibitor, PJ34. Human intestinal organoids, which are originated from ulcerative colitis (UC) patients, were exposed to pro-inflammatory cytokines (INFγ + TNFα) to induce intestinal inflammation, or coexposed to cytokines and PJ34. Results show that PARP1^−/−^ mice develop less severe colitis than WT mice, evidenced by a significant decrease in fecal and serum HMGB1, and, similarly, treating WT mice with PJ34 reduces the secreted HMGB1. The exposure of intestinal organoids to pro-inflammatory cytokines results in PARP1 activation and HMGB1 secretion; nevertheless, the co-exposure to PJ34, significantly reduces the release of HMGB1, improving inflammation and oxidative stress. Finally, HMGB1 release during inflammation is associated with its PARP1-induced PARylation in RAW264.7 cells. These findings offer novel evidence that PARP1 favors HMGB1 secretion in intestinal inflammation and suggest that impairing PARP1 might be a novel approach to manage IBD.

## 1. Introduction

Inflammatory bowel diseases (IBD), comprising Crohn’s disease (CD) and ulcerative colitis (UC), are chronic relapsing inflammatory disorders, for which incidence is increasing worldwide [1]. It is generally accepted that IBD are caused by an abnormal immune response to the gut microbiota in genetically susceptible subjects; however, although molecular mechanisms leading to IBD have been deeply investigated, a comprehensive understanding of the disease etiopathogenesis remains uncertain, hampering the development of effective therapeutic strategies. Indeed, the current therapies for IBD include a wide repertoire of drugs, including amino salicylates, glucocorticoids, immunomodulators and targeted biologicals [2,3,4], Nonetheless, the identification of new molecular targets for the development of additional treatment options is of high clinical priority.

High-mobility group box 1 (HMGB1) is a nonhistone DNA-binding protein, highly conserved in mammalian tissues, that is responsible for maintaining the structure of nucleosomes and regulating gene transcription [5,6,7]. When passively released by dying cells or actively secreted by activated immune cells and other cells in response to stress signals, HMGB1 is released in the extracellular environment and affects multiple inflammatory responses, acting as a danger-associated molecular pattern (DAMP) or prototypical alarmin [8,9,10]. Due to its pro-inflammatory and immunostimulatory properties, HMGB1 is widely known to contribute to the pathogenesis of several chronic inflammatory and autoimmune diseases, such as sepsis, lung conditions, autoimmune diseases, acute liver injury, cardiac injury, neuroinflammation and other inflammation-driven conditions [11,12,13,14,15,16,17,18,19]. Accordingly, we showed that HMGB1 is highly expressed in the inflamed intestinal tissues of CD and UC patients and that experimental colitis in mice is highly improved by inhibiting HMGB1 [20,21,22,23]. The secretion and release of HMGB1 is fine-tuned by a variety of factors, including the post-translational modification of its two nuclear localization sequences (NLS) [7,24,25]. In particular, acetylation of lysine in the NLS favors the HMGB1 shift to the cytoplasm, impairing its return to the nucleus [26,27].

Poly (ADP-ribose) polymerase 1 (PARP1) is an enzyme traditionally known to be responsible for DNA repair. Upon DNA damage, PARP1 is recruited to the damaged sites and catalyzes the synthesis of polymer termed poly (ADP-ribose) chains [28]. However, several evidences highlight that PARP1 and poly ADP-ribosylation (PARylation) are key players in different aspects of inflammation as well [29,30,31]. Interestingly, recent studies show that PARP1 interacts with HMGB1 and promotes its activation and release [32,33,34], facilitating acetylation levels via PARylation [35,36,37,38,39].

Hence, the aim of the present study was to explore the relationship between PARP1 and HMGB1 in controlling gut inflammation by using *PARP1*^−/−^ mice and human intestinal organoids (HIOs). We offer novel evidence that PARP1 favors HMGB1 secretion and suggest that impairing PARP1 activity should be a novel approach to manage IBD patients.

## 2. Results

### 2.1. PARP1^−/−^ Mice Develop Less Severe Colitis than Wild type (WT) Mice

*PARP1*^−/−^ and WT mice (five animals per group) were treated with 3% dextran sodium sulphate (DSS for 7 days to induce severe colitis that was confirmed by macroscopic (animal weight, colon length, and clinical score) and microscopic (histology) endpoints. Results showed that *PARP1*^−/−^ mice develop less severe colitis than WT mice, as shown by decreased weight loss (* *p* < 0.05), a higher clinical score value (* *p* < 0.05) and an increased colon length (** *p* < 0.01), (Figure 1A–C). Accordingly, the histology showed an improved morphology of the intestinal mucosa with a recovery of the villi structure, a lower inflammatory infiltrate and a significant reduction in the histological score (** *p* < 0.01) in *PARP1*^−/−^ (Figure 1D,E).

Overall, these data highlight that the absence of PARP1 is protective against the onset of murine colitis.

### 2.2. Extracellular HMGB1 Is Significantly Reduced in the Stools and Serum of PARP1^−/−^ Compared to WT Mice

After comparing the expression of HMGB1 in inflamed colon tissue in *PARP1*^−/−^ and WT mice, no difference was noted (Figure 2A). This finding was in agreement with our previous results that showed that intracellular HMGB1 does not vary between the total protein of tissues with different degrees of inflammation, nor between inflamed and normal tissue [20]. This occurs because the protein is abundantly expressed and localized in the nucleus of most cells, including intestinal epithelial cells, and because the quantity secreted into the cytoplasm following an inflammatory insult is not appreciable. The levels of secreted HMGB1 in the serum and stools, however, were found to be significantly decreased in *PARP1*^−/−^ compared to WT mice (Figure 2B,C).

This last result suggests a relationship between PARP1 and HMGB1.

### 2.3. PARP1-Inhibitor, PJ34, Significantly Decreases the Amount of HMGB1 in the Stools of DSS-Treated WT Mice and Restores the Architecture of Intestinal Mucosa

PJ34 is a specific inhibitor of PARP1. WT mice were treated with DSS to trigger inflammation or cotreated with DSS and PJ34 (10 mg/kg). Untreated mice were used as controls. Mice were sacrificed after 7 days. Interestingly, the cotreatment showed a significant reduction in the HMGB1 released in the stools of mice compared to the group treated only with DSS (Figure 3A). Moreover, the histological analysis revealed that after 7 days of cotreatment, PJ34 induced a clear recovery of the architectural structure of the gut wall, as also confirmed by the improved histological score (Figure 3B).

Mice treated with PJ34 also showed a recovery of colon length, although the recovery was not statistically significant. They did not, however, show a recovery of weight, probably due to the limited treatment time (Figure 3C,D).

This result confirms that PARP1 is also involved in murine gut inflammation via the regulation of HMGB1 release.

### 2.4. Exposure of HIOs to Pro-Inflammatory Cytokines (INFγ + TNFα) Results in PARP1 Activation and HMGB1 Secretion

To address the relationship between PARP1 and HMGB1, we set up cultures of HIOs using stem cells isolated from intestinal crypts of uninflamed areas of UC patients newly diagnosed.

HIOs were treated with a mix of proinflammatory cytokines (cytomix: INFγ + TNFα, 1:1) at two different concentrations, 250 ng/mL or 150 ng/mL, for 24 h (Figure 4A). Then, the activation of PARP1 and the protein expression of HMGB1 were analyzed. Results showed that the exposure to both doses of cytomix promoted the cleavage of PARP1 into the two isoforms (89 and 55 kDa), implying the activation of the protein, thus, to be more conservative, we used the lowest dose of cytomix (150 ng/mL) in following experiments. As expected, intracellular HMGB1 did not vary among organoids (Figure 4B), while HMGB1 released in the organoid lumen was increased (Figure 4C).

### 2.5. PARP1-Inhibitor, PJ34, Significantly Reduces the Release of HMGB1 in the Lumen of HIOs

HIOs were treated with cytomix (150 ng/mL) alone or in combination with PARP1 inhibitor PJ34 (100 µM). Results showed that PJ34 caused a significant decrease in HMGB1 secretion in the lumen of organoids (Figure 4D,E).

These features confirm in an HIO model that PARP1 regulates the release of HMGB1.

### 2.6. Inhibition of PARP1 Results in a Significant Decrease in Inflammation and Oxidative Stress in HIOs

HIOs were exposed to cytomix (150 ng/mL) alone or coexposed to cytomix and PJ34 (100 µM). Then, the mRNA expression of three pro-inflammatory cytokines, IL-1β, IL-8 and TNFα and of the stress oxidative enzyme, dual oxidase maturation factor 2 (DUOXA2), was analyzed. The results showed that PJ34 significantly reduced the expression levels of TNFα and of DUOXA2, but not of the other two cytokines (Figure 5).

This result highlights that PARP1 markedly contributes to raising inflammation and oxidative stress levels in the gut.

### 2.7. HMGB1 Release during Inflammation Is Associated with PARP1-Induced PARylation

The murine macrophage cell line RAW 264.7 was used to demonstrate that inflammation enhances the PARP1-mediated PARylation of HMGB1 causing an increased secretion of the protein. Cells were treated with the endotoxin LPS (0.5 µg/mL) with or without PJ34 (10 µM and 50 µM). LPS-induced inflammation resulted in a significant increase in HMGB1 PARylation and consequent protein release. The addition of PARP1-inhibitor PJ34 strongly reduced the amount of PARylated HMGB1, consequently decreasing HMGB1 release in cell supernatants (Figure 6).

This result confirms that PARP1 interacts with HMGB1 via PARylation and promotes its release.

## 3. Discussion

HMGB1 is a nuclear non-histone protein that is released from the cell in response to damage or stress, inducing an inflammatory response [40]. Many aspects of the mechanisms of action of HMGB1 and of its export and secretion through the cell membrane have been revealed, yet they still hold many uncertainties.

Recent evidence suggests that the export mechanisms of HMGB1 under stress conditions are controlled by nuclear effectors. PARP1, traditionally known for its role in DNA repair but for which a function as an inflammatory mediator has been recently reported, was shown to regulate HMGB1 cytoplasm translocation via its PARylation, resulting in the promotion of inflammation.

In this study, the relationship between HMGB1 and PARP1 in controlling intestinal inflammation has been explored. We found that *PARP1*^−/−^ mice show less severe colitis highlighted by improved clinical parameters as well as by significantly reduced amounts of serum and fecal HMGB1. To further confirm this finding, we used the PARP1-inhibitor PJ34 [41]. PJ34 has also demonstrated a widespread therapeutic effectiveness in suppressing the inflammatory response in colitis, autoimmune diabetes, uveitis, ischemia and cancer [42,43,44,45]. In our experiments, WT mice, cotreated with DSS to induce an acute colitis and PJ34 showed a significant decrease in fecal HMGB1. A histological analysis confirmed that inflammation was strongly reduced and tissue structure improved in PJ34-treated mice. This evidence suggests a role for PARP1 in controlling intestinal inflammation.

It is well known that the alarmin HMGB1 plays an important role in IBD pathogenesis [46]. In previous studies, we detected substantial amounts of HMGB1 in the fecal stream of patients with CD and UC, indicating that the protein is abundantly secreted by mucosal cells during gut inflammation to such an extent that fecal HMGB1 has been proposed as a robust noninvasive biomarker of clinical and subclinical intestinal inflammation [20,21,22,23]. In this study, we have also used HIOs originated from the colon crypts of newly diagnosed UC patients to investigate the role of PARP1 in the release of HMGB1. Organoids reduce the gap between monolayer cell culture and whole-organism environments, recapitulating the complex cellular organization seen in vivo. HIOs were isolated from uninflamed areas of UC patients and were treated with pro-inflammatory cytokines to trigger inflammation. Interestingly, secreted HMGB1, collected from the internal lumen of organoids, was significantly increased in inflamed compared to uninflamed organoids. Conversely, intracellular HMGB1 did not vary among organoids: this feature was not unexpected since HMGB1 is a nuclear constitutive protein and usually performs primary functions, such as modulating chromatin accessibility. HMGB1 is abundantly expressed in almost all eukaryotic cells, making it impossible to appreciate variations induced by the shift of the protein in the cytoplasmic compartment before its release in the extracellular environment following an inflammatory injury or other alarm signals.

Accordingly, the PJ34-mediated inhibition of PARP1 resulted in a significant decrease in HMGB1 secretion in the lumen of inflamed compared to uninflamed organoids, confirming the role of PARP1 in regulating HMGB1 release after inflammation.

To assess whether the PARP1-mediated impairment of HMGB1 secretion resulted in a reduction in inflammation, we analyzed the expression level of the inflammatory cytokine TNFα. As a late inflammatory mediator, HMGB1 responds to the early inflammatory mediator TNFα [47] and, in turn, helps to increase its production, thereby maintaining and prolonging inflammatory responses. Indeed, TNFα was significantly reduced after the exposure of organoids to PJ34. Moreover, the PARP1-mediated decrease in secreted HMGB1 caused a strong reduction in oxidative stress as well, as evidenced by the marked decrease in the oxidative stress marker DUOXA2. The strict link between chronic inflammation and the increase in oxidative stress is traditionally known. All these findings further demonstrate the role of PARP1 in promoting inflammation by acting on HMGB1 release.

Recently, it has been reported that the hyperacetylation of HMGB1 is related to protein release [24]. Interestingly, PARP1 was shown to increase the LPS-mediated HMGB1 acetylation and its subsequent secretion [34]. On the basis of this evidence, we aimed to deeply explore the relationship between HMGB1 and PARP1-mediated PARylation during inflammation. Thus, we treated murine macrophages with the bacterial endotoxin LPS, a major inducer of inflammation, and observed that, after the treatment, the levels of PARylated HMGB1 were significantly increased. Accordingly, PARP1 inhibition caused a strong decrease in HMGB1 PARylation resulting in a reduced alarmin release. This finding strengthens the view that, during inflammation, PARP1 directly interacts with HMGB1 causing its PARylation with subsequent activation. Therefore, we speculate that in colon tissue PARP1 improves the translocation of HMGB1 to the cytoplasm by raising its PARylation that, in turn, facilitates its acetylation and subsequent secretion.

A current challenge for the clinical management of inflammatory disorders, including IBD, is to develop innovative tools that specifically attenuate DAMP-mediated inflammatory responses. Accordingly, a number of strategies have been used to prevent HMGB1 release or to inhibit its activities [48]. However, a limited number of clinically efficacious inhibitors are currently available; therefore, fully understanding the molecular mechanisms involved in HMGB1 release will expand the therapeutic armamentarium in inflammatory diseases. The inhibition of PARP1 could represent a novel strategy to impair HMGB1 release and to improve inflammation. The increased understanding of this mechanism will enable the design of new drugs to neutralize HMGB1 and offer new perspectives for the resolution of complex disorders, such as IBD.

## 4. Materials and Methods

### 4.1. Animals

C57BL6/J WT and *PARP1*^−/−^ female mice (age: 8–9 weeks) were provided by Envigo (Milan, Italy) or born and hosted at the animal house of ENEA, respectively. Mice were housed in collective cages at 22 °C ± 1 °C under a 12 h light/dark cycle and with food and water ad libitum.

This study was conducted according to the European Community Council Directive 2010/63/EU, approved by the local Ethical Committee for Animal Experiments of the ENEA and authorized by the Italian Ministry of Health (n 76/2017-PR).

### 4.2. Animal Treatments

WT and *PARP1*^−/−^ mice (5 animals per group) were treated with 3% of dextran sodium sulphate (DSS; molecular mass, 36,000–50,000 Da, MP Biomedicals, Santa Ana, California) dissolved in drinking water ad libitum for 7 days to induce colitis. Furthermore, WT mice (9/10 animals per group) were untreated or treated with 3% DSS or cotreated with DSS 3% and 10 mg/kg/day of PARP1 inhibitor, [N-(6-oxo-5,6-dihydrophenanthridin-2-yl)-N,N-dimethylacetamide HCl] [41] (Selleckhem, Houston, TX, USA), which were diluted in PBS and administered twice daily by gavage.

Mice were checked daily for the clinical score by assessing the following parameters: behavior, body weight, stool consistency (0 for normal stool, 1 for moist/sticky stool, 2 for soft stool, and 3 for diarrhea), presence of blood in stools (0 for no blood, 1 for evidence of blood in stools or around anus, and 2 for severe bleeding), and general appearance of the animal (0 was assigned if normal, 1 for ruffled fur or altered gait, and 2 for lethargic or moribund), according to Maxwell et al. [49]. The percentage of weight loss was calculated in relation to the starting weight using the following formula: ([Weight on day X − Initial weight]/Initial weight) X 100. Stool specimens were collected at day seven and frozen at −80 °C before analysis. Mice were euthanized at the seventh day and the colon length was measured (from the anus to the top of the cecum). Distal colon samples were frozen in liquid nitrogen for RNA extraction or fixed for histology examination. Blood samples from submandibular vein were collected from each animal and sera were stored at −80 °C.

### 4.3. Histology

Distal colon samples were fixed in 10% formalin and embedded in paraffin for routine histology. Fixed colon tissues were transversally sectioned (4 µm thickness), mounted on glass slides, deparaffinized and stained using standard Hematoxylin and Eosin techniques. Sections were analyzed by light microscopy and scored according to the criteria of Maxwell et al. [49]. Experiments were carried out in a double-blind.

### 4.4. RT-PCR

Total RNA was extracted from mouse colonic tissues and collected organoids with the RNeasy kit (QiaGen GmbH, Hilden, Germany), and 1 µg of total RNA was reverse transcribed by a High-Capacity cDNA Reverse Transcription Kit (Applied Biosystems, Foster City, CA, USA) for colonic tissue and SuperScript™ III First-Strand Synthesis SuperMix (Invitrogen, by ThermoFisher Scientific, Monza, Italy) for HIOs.

Real-time PCR was carried out with an ABI PRISM 7300 Sequence Detection System using the SYBR Green kit (Applied Biosystems). GAPDH or RSP14, HPRT1 and B2M expressions were used to normalize target gene mRNA expression in murine colon tissue and HIOs, respectively. The quantity of mRNA relative to the reference gene was calculated by the 2^−ΔCT^ method.

Experiments were repeated 3 times. Primers used for real-time PCR are summarized in Table 1.

### 4.5. Quantification of Serum HMGB1 by Enzyme-Linked Immunosorbent Assay (ELISA)

HMGB1 was analyzed in the murine serum using the ELISA kit (MyBioSource from EMELCA Bioscience, Breda, The Netherlands); samples were diluted (1:200) in the kit-recommended diluent buffer.

### 4.6. Fecal Extraction

Murine stool specimens, stored at −80 °C, were resuspended in extraction buffer (ScheBo Biotech AG, Giessen, Germany) to a final concentration of 500 mg/mL. Samples were vortexed for 1 min at room temperature and placed in orbital shaking for 1 h at room temperature. After being centrifuged twice for 5 min at 10,000 rpm at 4 °C, clear supernatants were collected and stored at −80 °C. Total protein concentration was determined by the Bradford assay (Bio-Rad Laboratories, Hercules, CA, USA).

### 4.7. Patients

Mucosal biopsies from macroscopically noninflamed colon were obtained during routine endoscopy from 5 patients with a new diagnosis of UC at the Pediatric Gastroenterology and Liver Unit, Maternal and Child Health Department, Sapienza University of Rome, Policlinico Umberto I Hospital, Rome, Italy. Biopsy samples were collected in ice-cold basal medium (DMEM:F12 1:1 (Lonza, Basel, Switzerland) supplemented with 1X GlutaMAX (Gibco, ThermoFisher Scientific, Waltham, MA, USA), 10 mM HEPES (Gibco) and 100 U/mL + 100 µg/mL penicillin/streptomycin (Gibco), and processed within 2 h for crypt isolation.

All patients or caregivers gave written informed consent before sample collection (approved by the Ethics Committee of the Policlinico Umberto I Hospital, EC N°. 4771/2018).

### 4.8. HIOs

In order to have a reliable experimental model able to closely reproduce the complex spatial morphology of the intestinal epithelium, we set up HIO cultures.

For this purpose, intestinal crypts were isolated from 4 colon biopsies per individual following a previous protocol [50]. Biopsies were washed in chelating solution (CS) (distilled water with 5.6 mmol/L Na_2_HPO_4_, 8.0 mmol/L KH_2_PO_4_, 96.2 mmol/L NaCl, 1.6 mmol/L KCl, 43.4 mmol/L sucrose, 54.9 mmol/L D-sorbitol, and 0.5 mmol/L DL-dithiothreitol). Biopsies were incubated in CS containing 2 mmol/L EDTA for 45 min on a rocking platform at 4 °C. The medium (CS + EDTA) was then removed and fresh CS was added. Biopsies were then subjected to rigorous pipetting to create four crypt-containing fractions of 10 mL each, which were pooled after evaluation under a microscope. 10 mL of FBS was added and crypts were spun down at 1200 rpm (150–200 g). Supernatants were removed and crypts were resuspended in Matrigel (Growth Factor Reduced, phenol-red-free, Corning, NY, USA) diluted with basal medium (50/50%). To allow the formation of 3D organoid structures, four droplets of each cell suspension were plated in every well of 24 tissue culture plates. After polymerization, 500 µL of human expansion medium (HM) (WNT3A 50% *v*/*v* (in house, cell line), R-spondin-1 20% *v*/*v* (in house, cell line), Noggin 10% *v*/*v* (in house, cell line), EGF 50 ng/mL (Life Technologies, Carlsbad, CA, USA), A83-01 500 nM (Tocris Bioscience, Bristol, UK), SB202190 10 µM (Sigma-Aldrich, Saint Louis, MO, USA), Nicotinamide 10 mM (Sigma-Aldrich), n-Acetylcysteine 1.25 mM (Sigma-Aldrich), and B27 1X (Life Technologies)) was added to the wells. HIOs were refreshed every 2–3 days and split after 7–10 days.

### 4.9. HIOs Treatments

HIOs were placed in HM medium and exposed to a mix of pro-inflammatory cytokines (cytomix: INFγ + TNFα, 1:1) of 250 ng/mL or 150 ng/mL for 24 h to induce inflammation. Furthermore, HIOs were co-treated with cytomix (150 ng/mL) and PJ34 (100 µM) dissolved in DMSO for 24 h.

### 4.10. HIOs Protein Analysis

HIOs were mechanically disaggregated to allow the release of lumen content and centrifuged at 800 rpm for 10 min. Supernatants were recovered and Western blot assay was performed. Furthermore, organoids were suspended in ice-cold lysis buffer (50 mM Tris (pH 7.4), 5 mM EDTA, 250 mM NaCl, 0.1% Triton X-100, 1 mM phenylmethylsulfonyl fluoride, 5 mg/mL aprotinin, 5 mg/mL leupeptin, and 1 mM sodium orthovanadate (Sigma-Aldrich), and cell lysates were analyzed by Western blot.

Total protein concentration was determined by the Bradford assay (Bio-Rad Laboratories, Hercules, CA, USA).

### 4.11. Cell Culture

The murine macrophage-like cell line, RAW264.7, purchased from ATCC (Rockville, MD, USA), was cultured in RPMI 1640 medium, containing 10% fetal calf serum (FCS), 2 mM L–glutamine, 100 U/mL penicillin and 100 µg/mL streptomycin (Biochrom, Berlin, Germany), at 37 °C, 5% CO_2_. Cells were exposed to 0.5 µg/mL of the endotoxin lipopolysaccharides (LPS) (Sigma-Aldrich) to trigger inflammation or coexposed to 10 or 50 µM of PJ34 dissolved in PBS for 24 h. Proteins were extracted for immunoprecipitation and Western blot analyses.

### 4.12. Immunoprecipitation

RAW264.7 cells were lysed in lysis buffer (NTN) (120 mM NaCl, 50 mM Tris-HCl pH 8.0, and 0.1% Nonidet P-40) supplemented with 1 mM PMSF, 1 mM Na_4_VO_3_, 5 μg/mL leupeptin. Cell lysates were precleared with 100 μL protein A-Sefarose beads for 2 h at 4 °C and centrifuged at 200 rpm for 5 min. Supernatants were incubated with 2 µg of rabbit anti-HMGB1-Ab (Sigma-Aldrich) for 2 h at 4 °C. Protein A-Sepharose was added and incubated overnight at 4 °C. Beads were washed three times with NENT buffer (100 mM NaCl, 1 mM EDTA, 20 mM Tris-HCl pH 8.0, and 0.1% Nonidet P-40 supplemented with 1 mM PMSF, 1 mM Na_4_VO_3_, and 5 mg/mL leupeptin) centrifugated and collected for Western blot.

### 4.13. Western Blot

Protein extracts from mice (stools: 20 µg), HIOs (organoid cells: 10 µg for HMGB1 and 30 µg for PARP1; organoid lumen content: 25 µL) and RAW264.7 cells (5 µg) were analyzed.

Proteins were transferred in polyvinylidene fluoride membrane (Bio-Rad) and blocked with TBS-T (Tris-buffered saline with 0.1% Tween-20) containing 5% non-fat dry milk. Anti-HMGB1 (1:1000; R&D system, by Bio-Techne EMEA, Abingdon, UK), anti-PARP1 (1:1000; Cell Signaling, Milan, Italy), anti-Par (1:1000, Trevigen, Helgerman Ct, Gaithersburg, MD, USA) and anti-β-actin (1:5000; Sigma-Aldrich) antibodies were diluted in TBS-T containing 3% non-fat dry milk and incubated overnight at 4 °C. Membranes were washed in TBS-T, incubated for 1 h with horseradish peroxidase-conjugated secondary antibody (Santa Cruz Biotechnology Inc., Heidelberg, Germany), washed in TBS-T and developed with ECL-Plus (GE Healthcare, Europe GmbH, Freiburg, Germany). Densitometrical analysis of the blots was performed using the software ImageQuant (GE Healthcare).

### 4.14. Statistics

All statistical analyses were performed with GraphPad Prism Version 8.0 software (GraphPad Software, San Diego, CA, USA). For in vitro and in vivo experiments, the comparison between two groups was performed using Mann–Whitney U test. Data are presented as means +/− SD. Differences were noted as significant at * *p* < 0.05, ** *p* < 0.01 and *** *p* < 0.001.

## 5. Conclusions

In sum, after treatment with DSS, *PARP1*^−/−^ mice show less severe colitis compared to wild type mice, as evidenced by the significantly reduced amounts of serum and fecal HMGB1. Accordingly, the use of PARP1-inhibitor PJ34 causes a significant decrease in HMGB1 secretion in wild type mice after DSS treatment. Furthermore, the treatment of HIOs with pro-inflammatory cytokines results in PARP1 activation and HMGB1 secretion; meanwhile, PJ34 significantly reduces the release of HMGB1 in the lumen of HIOs. Interestingly, HMGB1 release during inflammation is associated with PARP1-induced PARylation in murine macrophages.

All these findings together highlight a role of PARP1 in controlling intestinal inflammation via HMGB1. We believe that PARP1-enhancing HMGB1 PARylation promotes its acetylation following its release in the extracellular environment.

## Figures and Tables

**Figure 1 ijms-24-07096-f001:**
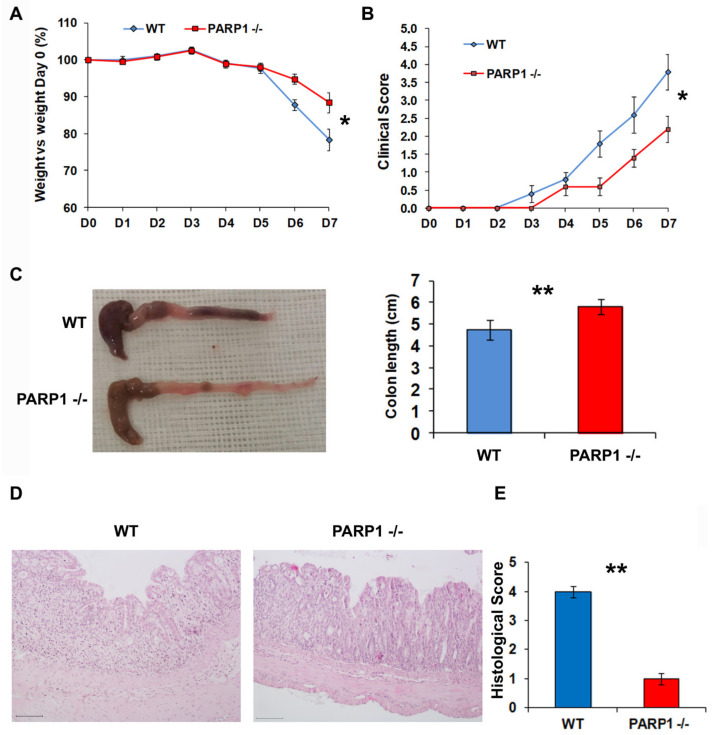
*PARP1*^−/−^ mice show less severe colitis than WT mice. *PARP1*^−^/^−^ and WT mice (5 animals per group) were treated with 3% DSS to induce acute colitis that was evidenced by analyzing (**A**) animal weight, (**B**) total animal clinical score, (**C**) animal colon length, (**D**) histology (Scale bar = 100 µm) and (**E**) histological score. Statistical analysis was performed using the Mann–Whitney U test. * *p* < 0.05; ** *p* < 0.01.

**Figure 2 ijms-24-07096-f002:**
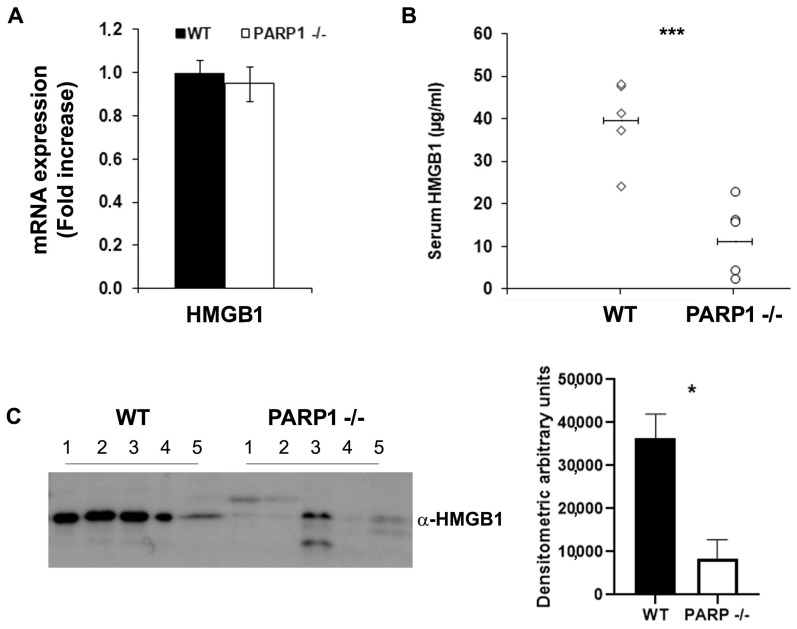
The release of HMGB1 in the stools and serum of *PARP1*^−/−^ mice is significantly decreased compared to WT mice (5 animals per group). Analysis of intracellular (**A**), serum (**B**) and fecal (**C**) HMGB1 in PARP1^−/−^ and WT mice treated with 3% DSS. Statistical analysis was performed using the Mann–Whitney U test. * *p* < 0.05; *** *p* < 0.001.

**Figure 3 ijms-24-07096-f003:**
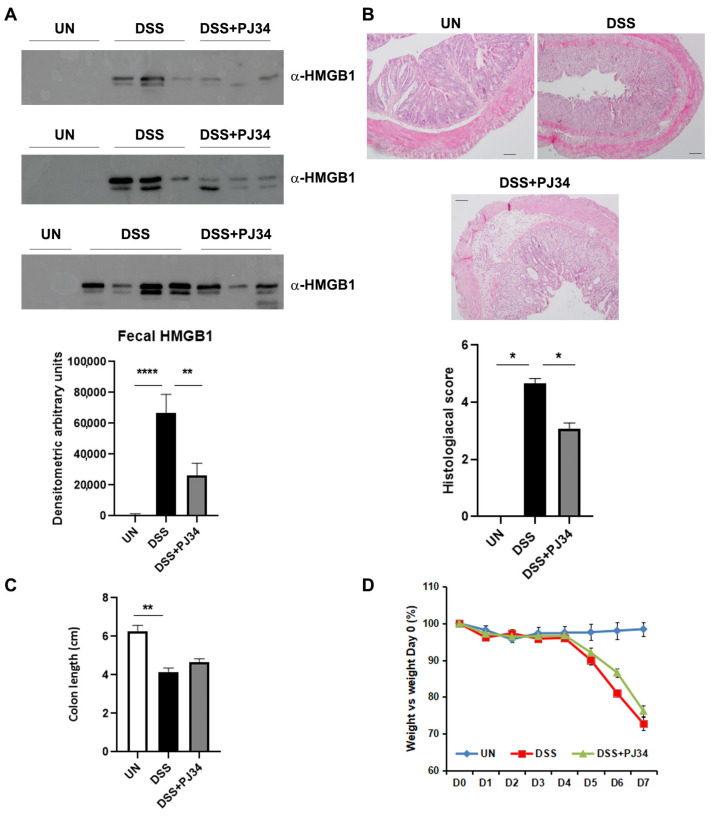
PARP1-inhibitor, PJ34, strongly reduces fecal HMGB1 in DSS-treated WT mice and improves the architecture of intestinal mucosa. WT mice were treated with 3% DSS (10 animals) or cotreated with DSS and 10 mg/kg of PARP1 inhibitor, PJ34 (9 animals). (**A**) Analysis of fecal HMGB1; (**B**) histology and histological score (Scale bar = 100 µm); (**C**) animal weight; (**D**) animal colon length. Statistical analysis was performed using the Mann–Whitney U test. * *p* < 0.05; ** *p* < 0.01; **** *p* < 0.0001.

**Figure 4 ijms-24-07096-f004:**
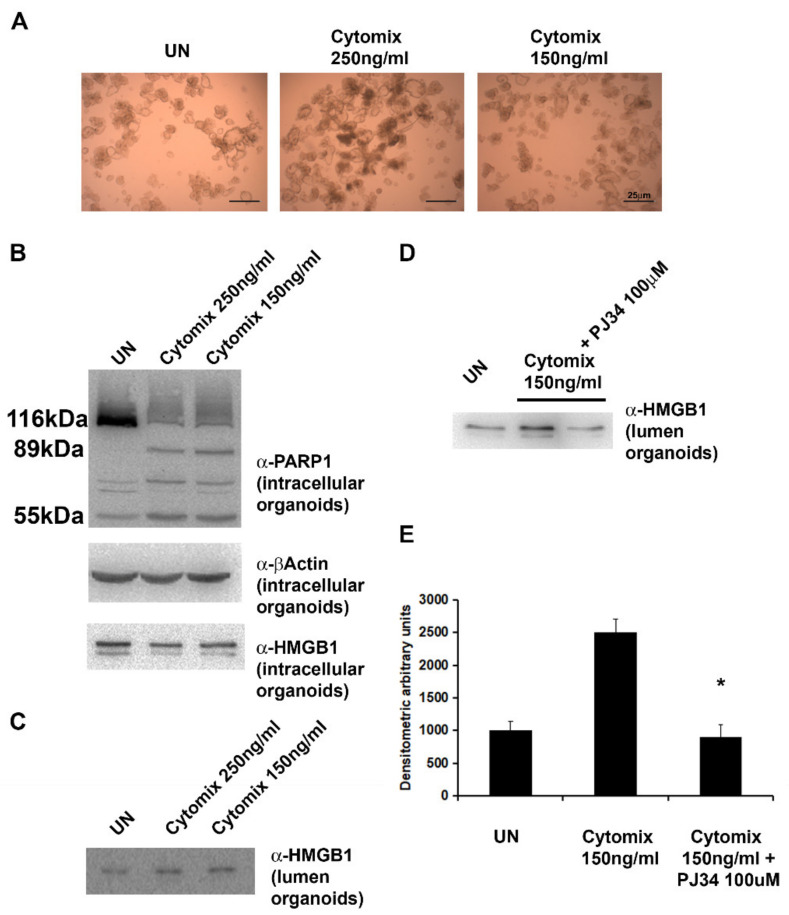
Pro-inflammatory cytokines INFγ and TNFα cause PARP1 activation and HMGB1 secretion in HIOs. The treatment with PJ34 significantly reduces the release of HMGB1 in the lumen of HIOs. HIOs were treated with cytomix or cotreated with cytomix and PJ34. (**A**) HIO morphology (Scale bar = 25 µm). Western blot of (**B**) PARP1 and intracellular HMGB1 or (**C**,**D**) secreted HMGB1. (**E**) Densitometric analysis of secreted HMGB1. The experiment was performed in triplicate. Statistical analysis was performed using the Mann–Whitney U test. * *p* < 0.05.

**Figure 5 ijms-24-07096-f005:**
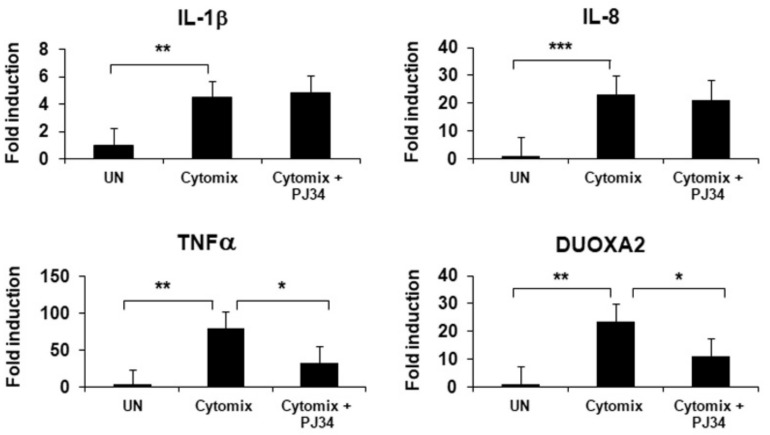
PJ34 treatment significantly improves inflammation and oxidative stress in HIOs. mRNA expression of pro-inflammatory cytokines IL-1β, IL-8 and TNFα and oxidative stress marker DUOXA2 in HIOs exposed to cytomix or coexposed to cytomix and PJ34. Statistical analysis was performed using Mann–Whitney U test. * *p* < 0.05; ** *p* < 0.01; *** *p* < 0.001.

**Figure 6 ijms-24-07096-f006:**
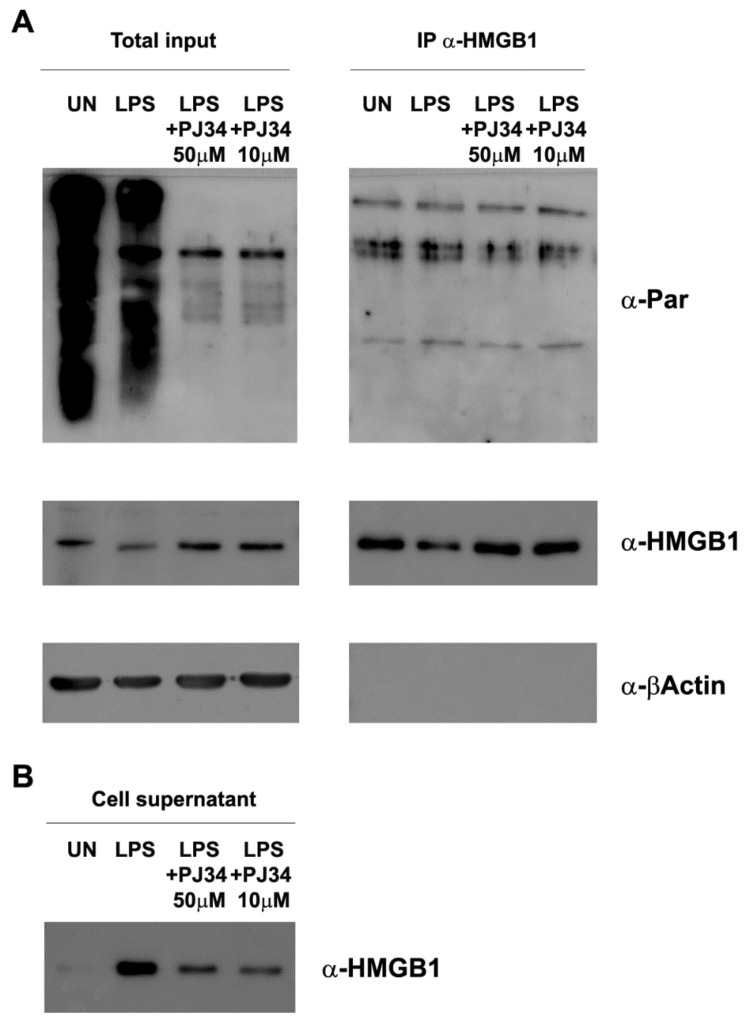
PARP1 favors HMGB1 PARylation promoting HMGB1 release. (**A**) Immunoprecipitation and Western blot of HMGB1 in RAW264.7 cells treated with LPS or cotreated with LPS and PJ34. Protein lysates were immunoprecipitated with anti-HMGB1 antibody followed by immunoblotting with anti-Par antibody, anti-HMGB1 or anti-βActin as a negative control of IP. (**B**) Western blot of HMGB1 in cell supernatant of RAW264.7 cells treated with LPS or cotreated with LPS and PJ34. The experiment was performed in duplicate.

**Table 1 ijms-24-07096-t001:** List of primers for real-time PCR.

Gene	Primer Forward	Primer Reverse
mHMGB1	GCCTCGCGGAGGAAAATC	AAGTTTGCACAAAGAATGCATATGA
mGAPDH	AACTTTGGCATTGTGGAAGG	CACATTGGGGGTAGGAACAC
hIL-1β	TTCGACACATGGGATAACGAGG	TTTTTGCTGTGAGTCCCGGAG
hIL-8	TTGGCAGCCTGATTTC	TTGGAGTATGTCTTTATGCACTGAC
hTNF-α	ATCTTCTCGAACCCCGAGTGA	GGAGCTGCCCCTCAGCTT
hDual Oxidase Maturation Factor 2 (Duoxa2)	TCCGCAACGATGGACAGA	AGGGTCGGTTGGAAACGAA
hRibosomal Protein S14 (RSP14)	TCACCGCCCTACACATCAAAC	GCCCGATCTTCATACCCGA
hHypoxanthine Phosphoribosyltransferase 1 (HPRT1)	GAAAAGGACCCCACGAAGTGT	AGTCAAGGGCATATCCTACAACA
hBeta-2-Microglobulin (B2M)	TGCTGTCTCCATGTTTGATGTATCT	TCTCTGCTCCCCACCTCTAAGT

## Data Availability

All data analyzed during this study are included in the manuscript.

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
