# Peer review of "PARP1 Activation Induces HMGB1 Secretion Promoting Intestinal Inflammation in Mice and Human Intestinal Organoids"

_ijms, 2023, doi:10.3390/ijms24087096_

Round 1

Reviewer 1 Report

see attached review comments

Reviewer 2 Report

Thank you for inviting me to evaluate the article titled “Poly (ADP-ribose) Polymerase (PARP)1 activation induces High Mobility Group Box 1 (HMGB1) secretion promoting intestinal inflammation in mice and human intestinal organoids". In this article, the authors investigated the relationship between HMGB1 and PARP1 in the control of intestinal inflammation. PARP1 was found to favor the production of HMGB1 in intestinal inflammation and suggests that weakening PARP1 may be a novel approach to managing IBD. The subject matter of this manuscript is interesting and the manuscript provides important results. Some comments have been included below and perhaps some additional comments and discussion by the author could make the manuscript writing clearer. The authors need to include more details about that. Based on the relevance of the subject matter and results, I recommend that the manuscript be accepted for entry into International Journal of Molecular Sciences after the authors have made significant changes to the manuscript.

1. It is recommended to use the abbreviations PARP1 and HMGB1 in the title.

2. The introductory section of the article needs to be streamlined, with relevant descriptions grouped into one paragraph.

3. In row 79, why are there 5 replicates and is there a small sample size.

4. Table 1 needs to be re-edited, not only because of the inconsistency in the font.

5. For abbreviated comments that only need to appear once, such as HIOs, please check the full text.

6. In line 189 pH8 need to add spaces in the middle, please check the full text.

7. In lines 80 and 202, the reagent information is incomplete, please check the full text.

8. For P-values that require italics, please check the full text.

9. Please explain why the Mann-Whitney U test was used.

10. There is an error in the significance marker of the histogram in Fig. 1; the staining color of Fig. 1D should be uniform.

11. Fig. 2C's need to indicate specific information about the protein, such as protein size as well as name, etc. Please check the full text.

12. Suggest adding sections for Graphical Abstract, Conclusions, Author Contributions, Funding, Institutional Review Board Statement:, Informed Consent Statement, Data Availability Statement. Acknowledgments, Conflicts of Interest, etc.

Reviewer 3 Report

This study explored the relationship between HMGB1 and PARP1 in controlling intestinal inflammation PARP1−/− mice showed a less severe colitis and significantly reduced amounts of serum and fecal HMGB1. Furthermore, the PARP1-inhibitor PJ34 has additionally shown a broad therapeutic efficacy in reducing the inflammatory response. When intestinal organoids were exposed to pro-inflammatory cytokines, PARP1 was activated and HMGB1 was secreted; however, when PJ34 was also present, the release of HMGB1 was greatly reduced, reducing inflammation and oxidative stress.

From a clinical perspective this study highlights the role of the inhibition of PARP1 as a new strategy to impair HMGB1 release and to improve inflammation. Indeed, further understanding of this mechanism might enable to design new drugs to neutralize HMGB1.

Major point of my critical appraisal is about the use of intestinal organoids.

1)      What it the rationale behind the use of organoids? This should be added in the methodology.

2)      In addition to this, authors stated in the discussion that organoids reduce the gap between monolayer cell culture and whole-organism environments, organoids replicate the intricate cellular organization seen in vivo, narrowing the gap between monolayer cell culture and whole-organism settings. HIOs were isolated from UC patients' non-inflamed regions, and pro-inflammatory cytokines were administered to them to cause inflammation. As a result, HMGB1 secretion in the lumen of inflamed compared to uninflamed organoids decreased significantly when PARP1 was inhibited by PJ34, demonstrating the importance of PARP1 in controlling HMGB1 release following inflammation. Are there any drawbacks in using these organoids compared to other more commonly used models (cellular components, standardizations of protocols, costs)? What is the microenvironment here?

Minor points

Figure 2,3,4 in WB actin control is not provided. Please re-do the triplicate of the WB or provide positive control. Furthermore, provide a quantitative analysis of the WB

Extensive copyediting is required in addition to English revision. (such as page 2 line 54 posttranslational, page 2 line 79 wilde)

Reviewer 4 Report

It has been reported that PARP deficiency protects DSS-induced intestinal inflammation (PMID: 26912654). So, the novelty of this article were the mechanism of PARP1-induced HMGB1 release and the usage of HIOs.

I think many problems are existed in this manuscript. The comments that I consider major or minor were as follows.

1.     Images in Figure 1D and 3D are indistinct, larger magnification or more clearly images should be presented.

2.     The size, type, and thickness of fonts, symbols, and lines in figures should be uniform, at least, not disorderly and unsystematic.

3.     All WB results need to use Internal Control Antibody. Only unification of loading protein is insufficient.

4.     The scale bar should be uniform in one figure. The manner of “ Scale bar 4X=200mM; 10X=100mM" is not accepted. Besides, the authors used lowercase m in Figure 1 BUT used capital M in Figure 3. These problems should also be checked carefully in all figures.

5.     The number for each group and statistical approach for each figure should be presented clearly in figure legends.

6.     In results 3.5, it indicates that PARP1 inhibition prevents HMGB1 release in cytokine-treated HIOs, suggested exogenous cytokine treatment induced an HMGB1 release. However, in 3.6, the authors measured the cytokines produced in HIOs. If the HIOs can produce cytokines, why they need exogenous cytokine treatment to induce the HMGB1 release. If not, what is the meaning of detecting cytokine mRNA levels in HIOs.

7.     The author mainly detected HMGB1 release in HIOs. However, in 3.7, they measured the relationship of HMGB1 release and PARP1-induced PARylation in macrophages. Why?

Round 2

Reviewer 3 Report

No further comments

Author Response

The authors really thank the Reviewer for the positive feedback

Reviewer 4 Report

Many of the concerns have been well addressed. But, considering the PARP1-induced HMGB1 release is the main novelty of this research, I still think the target cell subset should be clearly identified, and all the studies should tightly connected to the principal line.

For example, the authors can prove HMGB1 released from eukaryotic cells (or HIOs) is important, released by inflammatory stimuli, and the release cam further stimulated eukaryotic cells inflammatory in the absence of inflammatory cells. The HMGB1 release in eukaryotic cells (or HIOs) is associated with PARP1-induced PARylation. 

Or, the authors can prove HMGB1 released from macrophages is important. The HMGB1 release from macrophages, BUT NOT eukaryotic cells, contributes to  further stimulated eukaryotic cells inflammatory. The HMGB1 release in macrophages is associated with PARP1-induced PARylation. 

Many cells are also invovled in this model, like T cells, DCs, neutrophils. All of them can release HMGB1. So, the authors should identify the main target cells clearly, prove they are responsible, or at least partially responsible, for this disease model, instead of talking in generalities. Of note, the novelty of this article were only the mechanism of PARP1-induced HMGB1 release.

Round 3

Reviewer 4 Report

Considering the authors claimed that PARP1 favors HMGB1 secretion and it is the main novelty of this research, they should be provide more evidence to confirm the direct evidence of PARP1 to HMGB1, especially in one cell type. But not test HMGB1 release from many different cell types without any connection.

Besides, I do not have further concerns.

No further questions. 

Author Response

In fact we have evaluated the relationship between HMGB1 and PARP1 in multiple experimental models, however connected as cells of the immune system (even epithelial cells are now considered an integral part of the innate immune response) in the context of an immune-mediated disorder, since it seemed to us that this brought added value to our findings. However, we are very grateful to the Reviewer for this suggestion, we will certainly take it into consideration in the follow up of the study